# Core training elicits greater improvements than flexibility training in jumping lotus kick performance and physical attributes of Tai Chi athletes: A randomized controlled trial

Feng Li[2☯], Wenyan Yue[3☯], Hezhi Xie[4☯], Huiping Gong[1*]

1 Wushu School, Guangzhou Sport University, Guangzhou, P.R. China, 2 Graduate School, Guangzhou Sport University, Guangzhou, P.R. China, 3 School of Physical Education, Jinan University, Guangzhou, P.R. China, 4 School of Athletic Training, Guangzhou Sport University, Guangzhou, P.R. China

☯ These authors contributed equally to this work, are regarded as co-first authors.
* huipinggong97@gmail.com

## Abstract

Tai Chi's intricate, balance-intensive techniques, such as the Jumping Lotus Kick (JL Kick), demand both core stability and flexibility. Despite their importance, direct empirical comparisons of these training effects on Tai Chi performance remain limited. This study aimed to compare the impacts of an eight-week core stability regimen versus flexibility training on JL Kick proficiency and related physical attributes in Tai Chi athletes. In this randomized controlled trial, thirty-nine male athletes (ages 17–21) were allocated into core stability (n = 13), flexibility (n = 13), and control groups (n = 13). Performance metrics were assessed at baseline, midpoint, post-intervention, and at 12-week follow-up to capture both immediate and sustained effects. Core stability training yielded notable improvements in JL Kick rotation (ES = 1.2), core strength (Hanging Leg Raises, ES = 1.39), and dynamic balance (Y Balance Left, ES = 1.32), surpassing flexibility training, which primarily enhanced range of motion (Sit and Reach, ES = 1.44) without significant impact on JL Kick performance. Positive correlations between core strength, dynamic balance, and JL Kick performance (r = 0.45–0.68, p < 0.01) highlight core stability's essential role in Tai Chi proficiency. These findings support core stability training as a superior approach to flexibility training for advancing complex Tai Chi movements, with direct implications for optimizing athletic conditioning in Tai Chi and similar martial arts disciplines. These findings advocate for the inclusion of core stability-focused regimens to enhance performance outcomes in dynamic, balance-dependent techniques.

**Data availability statement:** All relevant data supporting the findings of this study are included within the manuscript and its Supporting Information files. The raw data underlying the graphs have been deposited in Figshare and are publicly available at https://doi.org/10.6084/m9.figshare.30520874.

**Funding:** The author(s) received no specific funding for this work.

**Competing interests:** The authors have declared that no competing interests exist.

## Introduction

Globally recognized for its health and wellness benefits, Tai Chi, or Taijiquan, has evolved beyond a meditative exercise into a discipline supporting key aspects of physical fitness, such as balance, flexibility, and coordination [1–4]. With increasing adoption in rehabilitation and fitness programs worldwide, Tai Chi's competitive dimension now demands significant physical prowess and specialized training to master intricate, high-performance techniques [5]. Among these, the Jumping Lotus Kick (téng kōng bǎi lián) exemplifies the rigorous physical demands of competitive Tai Chi, requiring substantial strength, agility, and precision [6,7]. This transformation highlights the need for research into targeted training approaches suited to the discipline's unique demands.

The Jumping Lotus Kick stands out as one of Tai Chi's most complex techniques, involving a mid-air rotation of 450° to 720° and precise hand-foot coordination, regulated by the International Wushu Federation (2024). Such technical requirements make the Jumping Lotus Kick an ideal focus for exploring the biomechanical and physical demands of advanced Tai Chi performance. The technique's rotational stability, control, and accuracy showcase the importance of specialized training methods.

Performing the Jumping Lotus Kick effectively depends on two core physical attributes: core stability and flexibility. Core stability facilitates efficient force transfer, balance, and control during high-speed rotations, essential for the precision demanded by this technique [8–11]. Flexibility, particularly in the hips and lower limbs, supports a range of motion critical for high kicks and multidirectional actions [12,13]. While static stretching improves ROM, reduces injury, and prevails in post-training martial arts, recent evidence favors dynamic flexibility for explosive neuromuscular control [13,14]. This distinction informs the comparison of flexibility and core stability training in optimizing Jumping Lotus Kick performance, remains understudied.

Although core stability and flexibility are well-studied across martial arts, their effects on Tai Chi's unique demands are less understood. Existing research in disciplines like Taekwondo and Karate addresses high-impact, linear actions, which differ from Tai Chi's fluid, rotational maneuvers [15,16]. While core stability is linked to balance and postural control [17], and flexibility aids multidirectional movements, their specific contributions to Tai Chi techniques involving sustained rotation remain unexplored. Closing these gaps is critical to developing evidence-based, discipline-specific training protocols for Tai Chi.

This study addresses this research gap by comparing the effects of core stability and flexibility training on Jumping Lotus Kick performance among male Tai Chi practitioners. We will assess the impacts on core strength, balance, reactive strength, and flexibility, which are hypothesized to enhance the technique's precision and effectiveness. By elucidating the differential effects of these training approaches, this research contributes essential insights for optimizing Tai Chi performance and refining training protocols for athletes and coaches.

## Methods

### Study design

This study employed a stratified randomized controlled trial to evaluate the effects of an 8-week core stability training program in comparison to a flexibility training program on Jumping Lotus Kick performance, core strength, flexibility, and balance among Tai Chi athletes. Three groups were included: a core training group (CTG), a flexibility training group (FTG), and a control group (CG). The study spanned 13 weeks, organized into distinct phases to ensure sufficient intervention exposure and accurate assessment. Specifically, the structure comprised a one-week familiarization period with baseline assessments, an 8-week intervention period, and a follow-up assessment at week 12. During the familiarization week, participants were introduced to the study's objectives, procedures, and the specific testing protocols to minimize the effects of learning and ensure protocol consistency. Baseline anthropometric and physical activity data were recorded to provide comprehensive initial metrics. Assessments were conducted at baseline, week 4, week 8, and during the follow-up at week 12. Each assessment was scheduled on weekends to ensure consistency with participants' training and minimize potential fatigue or interference from other weekly activities. The 8-week intervention period and its frequency of three sessions per week were selected to align with participants' typical training routines while allowing for adequate adaptation to the training stimuli without excessive risk of overtraining or injury. This choice is supported by prior research indicating significant adaptations to core and flexibility training over similar timeframes [10,18]. Each training session for the CTG and FTG groups was conducted separately from their usual Tai Chi sessions to ensure an independent assessment of the effects of each intervention.

### Participants

Sample size was calculated using G*Power (Version 3.1.9.7, University of Kiel, Germany) with $\alpha = 0.05$ and power $= 0.90$. The analysis indicated that 36 participants (12 per group) would be necessary to detect a large effect size ($f = 0.4$). To allow for potential attrition, the sample size was increased to 39, resulting in 13 participants per group. Of the 52 athletes initially assessed from the Guangzhou Sport University Wushu team, 13 did not meet eligibility criteria, and 39 were enrolled between March 18 and April 18, 2024.

Participants were male Tai Chi athletes aged 17–21 years, certified at Taijiquan Level 2 or higher by the Chinese Wushu Association, with a minimum of 5 years of continuous Wushu training and participation in at least three training sessions per week. Exclusion criteria included any musculoskeletal or orthopedic injury within the past 6 months, recent (within the past year) engagement in structured core or flexibility training, current medication affecting physical performance, and any physician-identified contraindications to exercise.

To ensure comparable baseline performance across groups, participants were stratified by initial Jumping Lotus Kick rotational degree before being randomly assigned to one of three groups: core training (CTG), flexibility training (FTG), or control (CG). An independent researcher conducted randomization using a computer-generated sequence and concealed group assignments in sequentially numbered, opaque envelopes, maintaining blinding until allocation. The study received ethical approval from the Guangzhou Sport University Academic Ethics Committee (approval number: 2024LCLL-99) and all participants were fully briefed on study procedures, risks, and benefits, and provided written informed consent. The individuals in this manuscript have given written informed consent (as outlined in PLOS consent form) to publish these case details.

### Training protocols

Participants in all groups completed their regular Tai Chi training sessions three times weekly on Monday, Wednesday, and Friday, each lasting 150 minutes. These sessions were led by certified Tai Chi instructors and included standardized warm-up, Tai Chi form practice, and cool-down. Following these sessions, participants in the core training group (CTG)

and flexibility training group (FTG) engaged in additional exercises tailored to their respective interventions, while the control group (CG) did not participate in supplementary training.

The CTG program consisted of core stability exercises designed to improve balance, control, and rotational strength essential for Tai Chi performance. Each session began with a 5-minute transition phase featuring Tai Chi breathing exercises to ease participants into supplementary training. The main exercises included three core stability exercises performed for 3 sets of 12–15 repetitions. These exercises included side plank with rotation, Bosu ball Tai Chi stance training, stability ball push-ups, and Single-Leg Balance with Weight Hold (see S1 Appendix for detailed exercise protocols). Exercise intensity was progressively increased every two weeks by adjusting repetitions or resistance [19]. A rest period of 1–2 minutes was provided between sets to support muscle recovery and balance strength-stability adaptations for Tai Chi demands [20,21] (S3 Fig).

To mirror conventional martial arts protocols, the FTG focused on static stretching post-Tai Chi, complementing the dynamic kicks inherent in the standard sessions shared across groups. Each session included a 5-minute transition phase with light Tai Chi stretches to prepare for supplementary flexibility exercises. The routine included 4–5 stretches per session, held 30–40 seconds at moderate discomfort for tolerability and 2–3 minutes cumulative exposure per group [22]. Stretching exercises included elevated front split, leg-elevated middle split, partner-assisted hip flexor stretches, and Partner-Assisted Hamstring Stretch (see S2 Appendix for detailed flexibility exercises and progression and S4 Fig). Progression occurred bi-weekly after the initial adaptation phase, with decisions guided by participant tolerance (e.g., absence of excessive fatigue) and coach observations per overload guidelines [22,23].

The control group (CG) adhered strictly to their regular Tai Chi practice sessions and refrained from additional physical activity. Weekly logs were used to confirm adherence, and any deviations were documented to ensure consistency within the group.

Both CTG and FTG interventions followed a bi-weekly progression schedule to encourage continuous adaptation, with the core group increasing repetitions or resistance and the flexibility group extending stretch durations—supported by evidence of ROM gains from gradual volume escalations over ≥2 weeks [24]. Session intensity was monitored via the Borg 0–10 RPE scale for the CTG only, recorded post-session for subjective workload [25].All supplementary sessions were conducted under the supervision of certified Tai Chi and strength conditioning coaches to ensure proper form, alignment, and safe intensity levels. Spotters were provided for balance-based exercises in the CTG to reduce injury risk. Participants were required to attend at least 80% of all sessions, with make-up sessions arranged for excused absences to maintain protocol adherence.

## Measures

**Anthropometry.** Participant height and body mass were assessed using an ultrasonic stadiometer and digital scale (DST-500, Beijing, China) with accuracy to ±0.5 cm for height and ±0.1 kg for body mass. Body mass index (BMI) was calculated by dividing body mass (kg) by the square of height ($m^2$) to obtain a value in $kg/m^2$.

**Jumping lotus kick rotational degree.** The rotational degree of the Jumping Lotus Kick was quantified using Kinovea software (v0.8.15), which calculated the rotational degree by defining the angle between lines drawn from the initial foot position at lift-off to the final landing position (S1 Fig). The motion was captured with four high-definition cameras (Sony FDR-AX43), positioned at 90° intervals around the performance area, ensuring comprehensive angle analysis. During the assessment, the success or failure of the performed movement was evaluated based on the criteria outlined in the Wushu Taolu Competition Rules and Judging Methods [5]. Criteria for a failed attempt included: taking more than four steps, failing to achieve lift-off, the hand not striking the foot with a crisp sound, or the leg not raised above shoulder height. To ensure objective scoring, three certified Wushu judges independently assessed each attempt, with a minimum of two judges required to confirm the outcome. Judges were blinded to group allocations, and each participant performed three attempts, resting two minute between each trial. The highest rotation degree from a successful attempt was selected for analysis.

 

**Reactive strength (rebound jump index).** Reactive strength and leg power were assessed using the Rebound Jump Test, which quantifies force production critical to the Jumping Lotus Kick [26]. The test was conducted on a contact mat (Smartjump, Fusion Sport), sampling at 1000 Hz, to capture both flight and contact times. Participants performed six maximal jumps, with instructions to minimize ground contact and maximize jump height. The Rebound Jump Index (RJI) was calculated as the ratio of flight time to contact time, following validated methods by Markovic et al (2004) [27]. Two trials were conducted per participant, with a 2-minute recovery interval between trials, and the highest RJI value was used in further analyses.

**Sit and reach test.** Flexibility was evaluated using the Sit and Reach Test, targeting hamstring and lower back flexibility. This assessment used a standardized sit-and-reach box (Baseline, White Plains, NY) with results recorded to the nearest 0.5 cm. Following a standardized 5-minute jogging and dynamic stretching warm-up, participants sat with legs extended and feet against the box, reaching forward along the scale. Each participant completed three attempts, holding each reach for 2 seconds, with the best result recorded. The reliability and validity of this protocol for flexibility assessment have been established in prior research [28].

**Hanging leg raises.** Core endurance was measured using the Hanging Leg Raise test, which quantifies abdominal strength and endurance. Participants hung from a pull-up bar with arms fully extended, using a pronated grip. The task was to raise the legs from a vertical position to shoulder height, maintaining a neutral spine throughout the motion. Repetitions were completed in a controlled manner without using momentum (S2 Fig), and the test was terminated upon form deterioration or volitional fatigue. The total number of correctly executed repetitions was recorded. This measure has demonstrated reliability in similar athletic contexts [29].

**Y-balance test (YBT).** The Y-Balance Test (YBT) assessed dynamic balance, core stability, and lower limb control, with measurements normalized to leg length as per standardized protocols. Conducted barefoot, participants performed single-leg reaches in anterior, posteromedial, and posterolateral directions while maintaining stance on the opposite leg. Leg length was measured from the anterior superior iliac spine to the medial malleolus. After three familiarization trials, each participant performed three recorded trials in each direction, with a 60-second rest interval between trials. The maximum reach distance in each direction was used for analysis. Research supports the reliability of the YBT in assessing dynamic balance [30].

## Measurement reliability

Test-retest reliability of the outcome measures was assessed in a separate sample of 20 Wushu athletes tested 1 week apart under identical conditions. Intraclass correlation coefficients (ICC) with 95% confidence intervals and coefficients of variation (CV) were calculated. ICC values >0.90 were considered excellent, 0.75–0.90 good, 0.50–0.75 moderate, and <0.50 poor [31]. CVs were deemed good (<5%), moderate (5–10%), or poor (>10%) [32,33]. Reliability data are presented in Table 1.

**Table 1. Test-retest reliability of outcome measures.**

| Outcome Measure | ICC(95% CI) | CV (%) |
| --- | --- | --- |
| Jumping Lotus Kick (°) | 0.908 (0.860, 0.950) | 4.6 |
| Core Strength (Reps) | 0.903 (0.850, 0.950) | 5.1 |
| Flexibility (Sit & Reach) | 0.924 (0.880, 0.960) | 3.9 |
| Balance (Y-Balance Test) | 0.946 (0.900, 0.970) | 4.2 |

High ICC values (>0.90) and low CV values (<5%) across all measures indicate excellent reliability, supporting the consistency of Tai Chi performance assessments and the validity of study results.

Reliability testing was conducted to ensure consistent and accurate outcome measures for Tai Chi performance, specifically for core strength, flexibility, balance, and rotational degree in the Jumping Lotus Kick. A pilot sample of 20 male Tai Chi athletes (ages 17–21) was tested under identical conditions to the main study, including standardized equipment, location, and time of day. Measurements followed the main study's protocols. Reliability metrics included the Intraclass Correlation Coefficient (ICC) for absolute agreement and Coefficient of Variation (CV) for measurement precision. According to Koo and Li (2016) [31], ICC values >0.75 indicate good reliability, and CV<5% [32,33] is considered excellent in Table 1.

## Statistical analysis

Data were presented as means ± standard deviations (SD). Normality of dependent variables was assessed using the Shapiro-Wilk test. A 3 (group: CTG, FTG, CG) × 4 (time: baseline, mid, post, follow-up) mixed ANOVA analyzed the effects of group and time on Tai Chi performance outcomes, including core strength, flexibility, balance, and Jumping Lotus Kick rotational degree. Group was treated as a between-subjects factor and time as a within-subjects factor. Greenhouse-Geisser corrections were applied for any sphericity violations. Significant main effects and interactions were further examined using post-hoc pairwise comparisons with Bonferroni correction. Effect sizes were reported as partial eta-squared ($\eta^2$) and interpreted per Cohen's (1988) criteria: small (0.01), medium (0.06), large (0.14) [34]. Pearson's or Spearman's correlation analyses assessed relationships between core strength and Tai Chi performance, with correlation strengths categorized as weak (r<0.3), moderate (r=0.3–0.5), or strong (r>0.5). The significance level was set at $\alpha \leq 0.05$. Analyses were conducted in R software (v4.3.3).

## Results

### Participant characteristics

A total of 39 male Wushu athletes (mean age: 19.3 ± 1.3 years; height: 169.3 ± 4.5 cm; weight: 65.2 ± 5.6 kg; training experience: 6.8 ± 1.1 years) completed the study. Baseline characteristics were similar across groups (Table 2).

### Intervention effects

**Jumping lotus kick rotational degree.** A significant Group × Time interaction was found for Jumping Lotus Kick rotational degree ($F_{(2.02, 36.35)} = 3178.98$, $p < 0.001$, $\eta^2 = 0.13$). Both CTG and FTG showed greater improvements than CG at post-intervention and follow-up, with CTG demonstrating the largest gains (Table 3, Fig 1A). Effect sizes

**Table 2. Participant characteristics by group.**

| Group | Age (years) | Height (cm) | Weight (kg) | Training (years) |
|---|---|---|---|---|
| FTG (n = 13) | 18.92 ± 1.5 | 168.7 ± 3.98 | 65.74 ± 5.08 | 7.19 ± 1.0 |
| CTG (n = 13) | 19.85 ± 1.07 | 168.75 ± 5.19 | 65.90 ± 7.63 | 6.9 ± 1.11 |
| CG (n = 13) | 19.23 ± 1.3 | 170.5 ± 4.37 | 64.07 ± 3.65 | 6.33 ± 1.12 |

Data presented as mean ± SD. FTG, flexibility training group; CTG, core training group; CG, control group.

**Table 3. Changes in jumping lotus kick rotational degree by group over time.**

| Group | Baseline (°) | Mid (°) | Post (°) | Follow-Up (°) |
|---|---|---|---|---|
| CG | 462.70 ± 13.72 | 464.67 ± 13.77 | 466.16 ± 13.82 | 465.16 ± 13.79 |
| CTG | 459.13 ± 23.01 | 481.09 ± 24.10*† | 508.78 ± 25.49*†‡ | 496.20 ± 24.79*†‡ |
| FTG | 459.29 ± 17.55 | 468.45 ± 17.90 | 481.17 ± 18.39* | 472.55 ± 18.06* |

Data presented as mean ± SD. * p<0.05 vs. baseline; † p<0.05 vs. FTG; ‡ p<0.05 vs. CG.

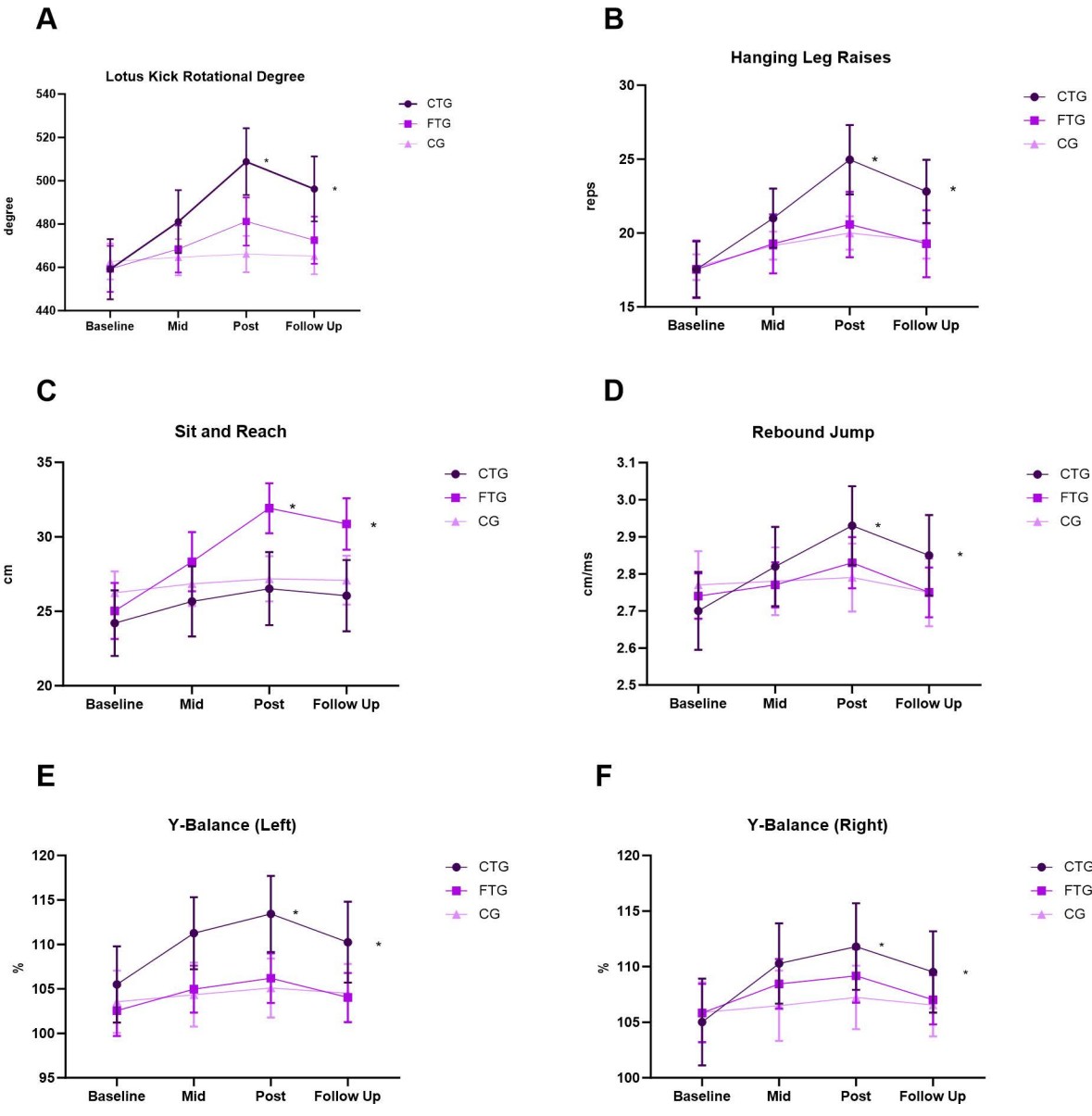

**Fig 1. Changes in athletic performance measures over time.** Data presented as mean with 95% CI. * p < 0.05 between groups.

(ES) and 95% confidence intervals (CI) for between-group comparisons at post-intervention were as follows: CTG vs. CG (ES = 2.16, 95% CI [1.43, 2.89]), FTG vs. CG (ES = 1.25, 95% CI [0.62, 1.88]), and CTG vs. FTG (ES = 0.91, 95% CI [0.31, 1.51]).

**Core strength (hanging leg raises).** A significant Group × Time interaction was found for Hanging Leg Raises (F(3.28, 59.06) = 74.24, p < 0.001, η² = 0.08). CTG significantly outperformed CG at post-intervention (ES = 1.39, 95% CI [0.62, 2.16]) and follow-up (ES = 0.99, 95% CI [0.25, 1.73]) (Table 4, Fig 1B). No significant differences were observed between FTG and CG.

**Table 4. Changes in secondary outcomes by group over time.**

| Outcome | Group | Baseline | Mid | Post | Follow-Up |
|---|---|---|---|---|---|
| Hanging Leg Raises (reps) | CG | 17.69±1.44 | 19.15±1.56 | 20.00±1.86 | 19.46±1.96 |
| | CTG | 17.54±3.14 | 21.00±3.32*† | 24.96±3.88*†‡ | 22.81±3.54*†‡ |
| | FTG | 17.54±3.22 | 19.27±3.31 | 20.58±3.66 | 19.27±3.75 |
| Sit & Reach (cm) | CG | 26.24±2.37 | 26.85±2.47 | 27.18±2.51 | 27.09±2.73 |
| | CTG | 24.20±3.65 | 25.66±3.90 | 26.52±4.06 | 26.05±3.95 |
| | FTG | 25.02±3.11 | 28.33±3.29*†‡ | 31.92±2.78*†‡ | 30.86±2.85*†‡ |
| Rebound Jump (cm/ms) | CG | 2.77±0.15 | 2.78±0.15 | 2.79±0.15 | 2.75±0.15 |
| | CTG | 2.70±0.17 | 2.82±0.18 | 2.93±0.18*‡ | 2.85±0.18* |
| | FTG | 2.74±0.10 | 2.75±0.10 | 2.84±0.11 | 2.76±0.11 |
| Y-Balance Left (%) | CG | 103.55±5.79 | 104.35±5.96 | 105.10±5.49 | 104.51±5.44 |
| | CTG | 105.05±7.10 | 111.26±6.70*†‡ | 113.42±7.09*†‡ | 110.25±7.52*†‡ |
| | FTG | 102.55±4.72 | 104.97±4.36 | 106.20±4.62 | 104.03±4.56 |
| Y-Balance Right (%) | CG | 105.88±4.48 | 106.48±5.24 | 107.23±4.72 | 106.55±4.68 |
| | CTG | 105.01±6.45 | 110.28±5.97*†‡ | 111.80±6.44*†‡ | 109.52±6.06*† |
| | FTG | 105.83±4.33 | 108.45±3.68 | 109.18±4.00 | 107.02±3.66 |

Data presented as mean±SD. * $p < 0.05$ vs. baseline; † $p < 0.05$ vs. FTG; ‡ $p < 0.05$ vs. CG.

**Sit and reach.** A significant Group×Time interaction was found for Sit and Reach ($F_{(2.92, 52.57)}$ = 78.84, $p < 0.001$, η²=0.09). FTG showed significantly greater improvements than CTG and CG at post-intervention (vs. CTG: ES=1.44, 95% CI [0.66, 2.22]; vs. CG: ES=1.71, 95% CI [0.90, 2.52]) and follow-up (vs. CTG: ES=1.33, 95% CI [0.56, 2.10]; vs. CG: ES=1.40, 95% CI [0.62, 2.18]) (Table 4, Fig 1C).

**Rebound jump index.** A significant Group×Time interaction was found for Rebound Jump Index ($F_{(2.56, 46.04)}$ = 120.75, $p < 0.001$, η²=0.06). CTG showed a significant improvement compared to CG at post-intervention (ES=0.80, 95% CI [0.06, 1.54]). No significant differences were observed between FTG and other groups (Table 4, Fig 1D).

**Y balance test.** Significant Group×Time interactions were found for the Y Balance test on both left ($F_{(2.94, 52.97)}$ = 21.34, $p < 0.001$, η²p=0.23) and right sides ($F_{(2.72, 49.02)}$ = 8.76, $p < 0.001$, η²p=0.15). CTG performed significantly better than other groups on the left side at post-intervention and follow-up (Table 4, Fig 1E-F). Effect sizes for between-group comparisons on the left side at post-intervention were: CTG vs. FTG (ES=1.07, 95% CI [0.32, 1.82]) and CTG vs. CG (ES=1.32, 95% CI [0.55, 2.09]). On the right side, CTG outperformed CG at post-intervention (ES=0.77, 95% CI [0.04, 1.50]).

## Correlations between variables

Significant positive correlations were observed between Jumping Lotus Kick performance and Hanging Leg Raises (r=0.45, $p < 0.05$), Rebound Jump (r=0.59, $p < 0.05$), Y Balance Left (r=0.41, $p < 0.05$) and Y Balance Right (r=0.23, $p < 0.05$), suggesting potential associations between core strength, lower limb power, balance and Jumping Lotus Kick performance (Table 5).

## Discussion

In light of emerging evidence on the role of core stability versus flexibility in athletic performance, this study sought to clarify whether core stability training produces more pronounced enhancements in Jumping Lotus Kick performance and related physical attributes, particularly core strength, balance, and reactive strength, compared to flexibility training. The results suggest that core stability training elicits superior improvements across these domains, underscoring the unique

**Table 5. Correlations between performance variables.**

|  | Jumping Lotus Kick (°) | Leg Raises (reps) | RJI(cm/ms) | Sit & Reach (cm) | Y-Balance Left (%) | Y-Balance Right (%) |
|---|---|---|---|---|---|---|
| Jumping Lotus Kick (°) | 1 | 0.45* | 0.59* | 0.21* | 0.41* | 0.23* |
| Leg Raises (reps) | 0.45* | 1 | 0.28* | 0.22* | 0.42* | 0.24* |
| RJI(cm/ms) | 0.59* | 0.28* | 1 | 0.29* | 0.01 | 0.17* |
| Sit & Reach (cm) | 0.21* | 0.22* | 0.29* | 1 | 0.17* | 0.10 |
| Y-Balance Left (%) | 0.41* | 0.42* | 0.01 | 0.17* | 1 | 0.03 |
| Y-Balance Right (%) | 0.23* | 0.24* | 0.17* | 0.10 | 0.03 | 1 |

* $p < 0.05$.

role of core stability in refining complex, rotational techniques in Wushu. In this regard, the findings contribute to a growing discourse challenging traditional flexibility-centered conditioning approaches in martial arts and position core stability as an essential component for technical mastery in disciplines emphasizing control, balance, and precision.

In the context of Jumping Lotus Kick rotational degree, the core stability group (CTG) exhibited substantial improvements over both the flexibility group (FTG) and control group (CG), with an effect size of 2.16 over CG. These findings align with the theoretical framework suggesting that core stability contributes to controlled force generation and transfer—a critical element in complex martial arts techniques that require rotational accuracy [10,11,35,36]. Core stability training, distinct from flexibility exercises, engages deep musculature that facilitates not only range of motion but also coordinated control over that range, essential for techniques demanding rapid, controlled rotation [37]. The literature on dynamic sports similarly emphasizes core stability's role in optimizing energy transfer along the kinetic chain, thereby enhancing the athlete's ability to regulate rotational force [17]. By consolidating stability and power, core stability training may better support performance in techniques requiring rotational control and accuracy, potentially positioning it as a more effective conditioning focus than flexibility alone.

Core strength, as evidenced by improvements in Hanging Leg Raises, demonstrated significant gains in CTG relative to FTG and CG, with a substantial effect size (ES = 1.39 over CG at post-intervention). Moreover, the positive correlation between Jumping Lotus Kick proficiency and core strength (r = 0.45, p < 0.05) suggests that a robust core significantly influences athletes' capacity to execute techniques requiring intense rotational control [10]. Mechanistically, core stability may enhance neuromuscular engagement within the core musculature, enabling greater postural rigidity and reducing energy dissipation during dynamic movements [38]. This process aligns with theories in motor control, which posit that core musculature functions as a stabilizing force that facilitates energy conservation and biomechanical efficiency, particularly during high-intensity movements [9]. Thus, in martial arts contexts, core stability likely acts as a foundational determinant of performance, ensuring both stability and efficiency in complex techniques like the Jumping Lotus Kick.

Regarding balance, the CTG showed marked improvement, especially on the left side (jumping leg), outperforming both FTG and CG with effect sizes of 1.07 and 1.32, respectively, over these groups at post-intervention. Core stability training enhances neuromuscular balance control, a critical factor for rotational shifts, though often conflated with proprioception, which involves sensory integration [38]. The observed correlation between balance and Jumping Lotus Kick proficiency (left side: r = 0.41, right side: r = 0.23) further highlights the relevance of core stability for maintaining equilibrium in high-accuracy techniques. The role of proprioceptive adaptation is well-documented in balance-intensive sports, with studies indicating that core training optimally activates neuromuscular pathways essential for spatial awareness and controlled movement [39]. This study suggests that core stability interventions promote proprioceptive gains that flexibility training alone cannot provide [40,41], ultimately fostering performance improvements in complex, precision-based techniques such as the Jumping Lotus Kick.

In assessing reactive strength through the rebound jump index, CTG participants displayed notable gains compared to CG, with an effect size of 0.80. The correlation between Jumping Lotus Kick performance and reactive strength ($r = 0.59$, $p < 0.05$) suggests that core stability training supports rapid power generation and controlled deceleration—key elements for executing explosive, high-energy techniques. Studies indicate that reactive strength improvements are associated with heightened motor unit recruitment and quicker muscle response times [9]. Core stability training may facilitate this neuro-muscular responsiveness, enhancing the athlete's capacity for quick stabilization and energy absorption during dynamic maneuvers [42–44]. As such, core stability training emerges as an effective conditioning strategy for techniques that demand a blend of rapid power adjustments and precise control.

These findings challenge the traditionally flexibility-focused conditioning paradigm in martial arts, highlighting core stability's unique contributions to rotational control and balance—a dimension underrepresented in flexibility training. This is particularly evident given static stretching's acute reductions in power and jump performance, which likely limited FTG's explosive outcomes, underscoring the need for dynamic approaches [45,46]. Flexibility has long been considered paramount for extending the range of motion, especially in Wushu. However, the present study's findings underscore that core stability, which supports balance and rotational control, may be equally, if not more, beneficial for complex techniques requiring multidirectional stability [47,48]. This study aligns with the broader athletic conditioning literature, which empha-sizes core stability's role in enhancing balance and rotational movements, thereby prompting a re-evaluation of flexibility's centrality in martial arts training [37,38]. The present study thus advocates a shift towards core-centered conditioning as a means of optimizing technical proficiency in Wushu and similar martial arts.

While this study provides compelling evidence for core stability training's benefits, certain methodological limitations warrant consideration. The limited sample size, primarily composed of male Wushu athletes, restricts the generalizability of these findings to other populations, including female athletes and those engaged in different martial arts disciplines. Future studies should expand sample diversity, investigating core stability's efficacy across a range of skill levels, age groups, and martial arts backgrounds to better understand its general applicability.

Moreover, despite substantial improvements observed in performance metrics, the study's design did not incorporate direct biomechanical analyses, such as electromyography or force vector measurement, to explore the specific neuro-muscular mechanisms behind the gains. Future research might employ biomechanical tools and RPE scales to enhance monitoring of FTG stretch intensity and CTG endurance, ensuring protocol-specific data capture [24,49].

Practical Applications for Training These findings carry important practical implications for coaches and athletes in Wushu and related martial arts. The observed benefits of core stability training on rotational control and balance suggest that training programs should prioritize exercises targeting these areas, such as planks, Russian twists, and stability ball exercises. Core stability training not only bolsters core strength but also enhances neuromuscular coordination, yield-ing improvements in balance and reactive strength crucial for performance optimization in complex techniques. These practical insights may extend to various sports beyond Wushu, particularly those requiring precise balance and rotational control, including gymnastics and figure skating.

Theoretical Contributions Theoretically, this study contributes to an evolving understanding of sports-specific condition-ing, suggesting that core stability, rather than flexibility, may be a critical factor for performance in balance-intensive and rotationally demanding sports. This paradigm shift aligns with emerging perspectives in motor control and skill acquisition, wherein core stability is viewed as a prerequisite for mastering complex movements in high-skill sports. These findings prompt a reevaluation of training frameworks that have traditionally prioritized flexibility, advocating instead for an inte-grated approach that considers core stability as foundational to performance in martial arts.

## Conclusion

In conclusion, the present study underscores the superiority of core stability training over flexibility training for improving Jumping Lotus Kick performance and related physical attributes in male Wushu athletes. By highlighting the significance

of core strength, balance, and reactive strength, these results advocate for a comprehensive approach to martial arts conditioning, challenging conventional flexibility-centric training models. Future research should build on these insights by examining core stability's biomechanical contributions and broadening the demographic scope of study populations. By integrating core stability training into conditioning programs, practitioners in martial arts and other dynamic sports may unlock greater potential for skill mastery and competitive success, ultimately redefining the conditioning practices in high-skill disciplines.

## Supporting information

**S1 Fig. Sequential phases of the jumping lotus kick (A-E).** The figure depicts the progression through each phase of the Jumping Lotus Kick, highlighting the initial rotational angle (−12.5°) and final rotational angle (−115.2°) as quantified using Kinovea software.
(TIF)

**S2 Fig. Hanging leg raise positions (A-C).** (A)Starting position, (B) Intermediate horizontal and (C) Final raised position.
(TIF)

**S3 Fig. Core training exercises (A-D).** (A) Side plank with rotation, (B) Bosu ball balance, (C) Stability ball push-ups and (D) Single-leg balance with weight.
(TIF)

**S4 Fig. Flexibility training stretches (A-D).** (A) Elevated front split, (B) Leg elevated middle split, (C) Partner-Assisted Hip Flexor Stretch and (D) Partner-Assisted Hamstring Stretch.
(TIF)

**S1 Appendix. Comprehensive protocol for core training: Exercise details and progression strategy.** This appendix provides a detailed breakdown of the core training exercises, sets, repetitions per side, rest intervals, and progression framework across an 8-week regimen aimed at systematically improving core stability and dynamic balance.
(DOCX)

**S2 Appendix. Comprehensive protocol for flexibility training: Exercise details and progression strategy.** This appendix provides a detailed breakdown of the exercises, sets, duration per side, rest intervals, and progression framework across an 8-week flexibility training regimen aimed at systematically improving range of motion and dynamic flexibility.
(DOCX)

## Acknowledgments

We are grateful for the assistance of the students and faculty of the Wushu school and the Sports Performance Laboratory of Guangzhou Sport University. The datasets generated and analyzed during the current study are not publicly available, but are available from the corresponding author who was an organizer of the study. The trial comply with the current laws of the country where they were performed.

## Author contributions

**Conceptualization:** Feng Li, Huiping Gong.

**Data curation:** Hezhi Xie.

**Formal analysis:** Feng Li, Hezhi Xie.

**Investigation:** Huiping Gong.

**Methodology:** Feng Li, Wenyan Yue, Hezhi Xie.

**Validation:** Wenyan Yue, Huiping Gong.

**Writing – original draft:** Feng Li, Hezhi Xie.

**Writing – review & editing:** Feng Li, Wenyan Yue, Huiping Gong, Hezhi Xie.

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
