## [Decision Letter · Decision Letter 0]

17 Jul 2025

Dear Dr. Gong,

We look forward to receiving your revised manuscript.

Kind regards,

Gianpiero Greco

Academic Editor

PLOS ONE

Journal Requirements:

2. We note that Supporting Figures S1 to S4 includes an image of a participant in the study. 

As per the PLOS ONE policy (http://journals.plos.org/plosone/s/submission-guidelines#loc-human-subjects-research) on papers that include identifying, or potentially identifying, information, the individual(s) or parent(s)/guardian(s) must be informed of the terms of the PLOS open-access (CC-BY) license and provide specific permission for publication of these details under the terms of this license. 

Please download the Consent Form for Publication in a PLOS Journal (http://journals.plos.org/plosone/s/file?id=8ce6/plos-consent-form-english.pdf). 

The signed consent form should not be submitted with the manuscript, but should be securely filed in the individual's case notes. Please amend the methods section and ethics statement of the manuscript to explicitly state that the patient/participant has provided consent for publication: “The individual in this manuscript has given written informed consent (as outlined in PLOS consent form) to publish these case details”. 

**Additional Editor Comments:**

Dear authors,

please reply point by point to the reviewers comments.

Reviewers' comments:

Reviewer's Responses to Questions

**Comments to the Author**

1. Is the manuscript technically sound, and do the data support the conclusions?

Reviewer #1: Partly

2. Has the statistical analysis been performed appropriately and rigorously?

Reviewer #1: Yes

3. Have the authors made all data underlying the findings in their manuscript fully available?

Reviewer #1: Yes

4. Is the manuscript presented in an intelligible fashion and written in standard English?

Reviewer #1: Yes

Reviewer #1: Review of the manuscript PONE-D-24-44478, entitled " Core Training Elicits Greater Improvements Than Flexibility Training in Jumping Lotus Kick Performance and Physical Attributes of Tai Chi Athletes: A Randomized

Controlled Trial"

The article presents a study aimed to compare the impacts of an eight-week core stability regimen versus flexibility training on JL Kick proficiency and related physical attributes in Tai Chi athletes.

This study highlights the superior effectiveness of core stability training over flexibility training in improving Jumping Lotus Kick performance and related physical attributes among male Wushu athletes. By emphasizing the importance of core strength, balance, and reactive strength, the findings advocate for a more comprehensive approach to martial arts conditioning, challenging traditional flexibility-focused training models. By incorporating core stability training into conditioning regimens, martial artists and athletes in other dynamic sports may unlock greater potential for skill development and competitive success, ultimately reshaping the conditioning strategies in high-performance disciplines.

The study presents the results of original research. The introduction section is well presented although references 14, 17, 42, 43 had to be edited as the title is in capital letters. Maybe could be added some extra recent investigation about the effects of core training on flexibility in other sports which requires also high bouts of dynamic stretching.

Study design, participants, training protocol, experiments, statistics, and other analyses are performed to a high technical standard and are described in sufficient detail. Conclusions are presented in an appropriate fashion. The article is presented in an intelligible fashion and is written in standard English.

In my opinion, specific minor changes must to be amended and considered before acceptance.

Comment #1: lines 49-50.

The authors state “While static stretching traditionally improves range of motion and mitigates injury risk, recent findings highlight the need for dynamic flexibility, aiding in neuromuscular control during explosive movements”, nevertheless their purpose for the FTG is based in static stretching. Please justify why did you choose static stretching if you already knew that is less efficient to improve explosive sports performance as the Jumping Lotus Kick?

Comment #2: Lines 130-139

The authors state: “The main exercises included three core stability exercises performed for 3 sets of 12-15 repetitions” and “A rest period of 1-2 minutes was provided between sets to support muscle recovery”. Normally, core stability programs aimed to improve the resistance of the core muscles. The purposed sets and repetitions are into the range of sets and repetitions included in resistance adaptations, nevertheless, the resting periods should be lower than 60” to avoid total recovery of core muscles and thus improve their adaptation to core stability. Despite of this, there were positive adaptations to this setup. Maybe in future research the authors should consider to purpose more challenging resting periods.

Comment #3: lines 143-144

The authors state:” The main stretching routine consisted of 4-5 stretches per session, held for 30-40 seconds each”. I miss some information about the amount of stretching tension supported by the participants as 30-40 sec is reported to be not enough to produce any effect on hamstrings elongation. Why don’t you apply longer stretching times? As is reported that stretching effects are better when the exposure times are longer, more than doing more than one set of shorter stretches.

Comment #4: 147-149

The authors state “Intensity was progressively increased bi-weekly by extending hold times or using assisted techniques, as needed. Participants rested for 30 seconds between stretches to prevent fatigue”. When did you applied assisted techniques or increasing the time? Justify the decision.

Moreover in lines

Comment #5: lines 155-157

The authors state: “The core group progressed through increased

repetitions or resistance, while the flexibility group extended stretch durations”. Could you justify by previous studies the chosen method to increase the time exposure in stretches? Was Borg 0-10 scale also be used for stretches?

Did you consider any method to assess athletes’ perception of stretch intensity?

Comment #6: line 270

Table format should be improved.

Comment #7: line 283

Table format should be improved.

Comment #8: line 313

Table format should be improved.

Comment #9: line 326

Table format should be improved.

Comment #10: line 337-341

The authors state “findings contribute to a growing discourse challenging traditional flexibility-centered conditioning approaches in martial arts and position core stability as an essential component for technical mastery in disciplines emphasizing control, balance, and precision.” You should express the limitations on doing static stretches, as is highly demonstrated that static stretches reduce power and jump performance, so needed in martial arts.

Coment #11: lines 372-374

The authors state: “This improvement suggests that core stability training directly enhances proprioceptive control, a critical factor in executing movements that require balanced rotational shifts”. Not all core stability exercises have this effect. Proprioception highly depends on imagery, focusing attention and reduce other afferent inputs that can negative interfere in the proprioceptive signal entry. Be careful on comparing proprioception with balance performance, as doing balance exercise not always improve the proprioceptive sense.

**Do you want your identity to be public for this peer review?** For information about this choice, including consent withdrawal, please see our Privacy Policy

Reviewer #1: **Yes: ** Monica Solana-Tramunt

---

## [Author Response · Author response to Decision Letter 1]

21 Aug 2025

Response to Reviewer Comments

Manuscript Title: Core Training Elicits Greater Improvements Than Flexibility Training in Jumping Lotus Kick Performance and Physical Attributes of Tai Chi Athletes: A Randomized Controlled Trial

Manuscript ID: PONE-D-24-44478

We would like to thank the Academic Editor and the reviewer for their constructive comments and valuable suggestions. We have revised our manuscript accordingly and provide detailed responses to each point below. All modifications have been marked in the revised manuscript with track changes.

Reviewer #1 Comments:

Comment 1:

Lines 49–50. The authors state: “While static stretching traditionally improves range of motion and mitigates injury risk, recent findings highlight the need for dynamic flexibility, aiding in neuromuscular control during explosive movements”, nevertheless their purpose for the FTG is based in static stretching. Please justify why did you choose static stretching if you already knew that is less efficient to improve explosive sports performance as the Jumping Lotus Kick?

Response:

We sincerely appreciate the reviewer's insightful observation regarding the potential discrepancy between our acknowledgment in the introduction of dynamic flexibility's benefits for neuromuscular control in explosive movements and the static-dominant protocol in the flexibility training group (FTG). We acknowledge that this rationale requires clearer articulation to align with the cited literature.

The FTG was designed to reflect traditional Tai Chi and Wushu practices, where static stretching post-training enhances range of motion (ROM) and reduces injury risk, as demonstrated by Bandy et al. (1998) in hamstring flexibility studies. Dynamic elements, such as kicks, were integrated into routine sessions, allowing FTG to leverage warmed muscles and avoid acute decrements (Behm et al., 2016). This enabled a valid comparator, yielding ROM gains (Sit and Reach, ES=1.44) without hindering Jumping Lotus Kick performance.

To enhance transparency, we propose revising the introduction (Lines 48–50) to: "While static stretching improves ROM, reduces injury, and prevails in post-training martial arts, recent evidence favors dynamic flexibility for explosive neuromuscular control(13,14) " .

In methods (Lines 140–142) add: "To mirror conventional martial arts protocols, the FTG focused on static stretching post-Tai Chi, complementing the dynamic kicks inherent in the standard sessions shared across groups".

Comment 2:

Lines 130–139. The authors state: “The main exercises included three core stability exercises performed for 3 sets of 12-15 repetitions” and “A rest period of 1-2 minutes was provided between sets to support muscle recovery”. Normally, core stability programs aimed to improve the resistance of the core muscles. The purposed sets and repetitions are into the range of sets and repetitions included in resistance adaptations, nevertheless, the resting periods should be lower than 60” to avoid total recovery of core muscles and thus improve their adaptation to core stability. Maybe in future research the authors should consider to purpose more challenging resting periods.

Response:

We thank the reviewer for their insightful comment on the 1-2 minute rest intervals in the core training group (CTG) protocol. These intervals may favor strength adaptations over endurance. This could be less optimal for Tai Chi's sustained rotational demands. Shorter intervals (<60 seconds) could induce greater metabolic stress. The chosen intervals aimed to promote ATP-CP resynthesis. This ensures high-quality repetitions vital for Tai Chi's postural precision and control (Willardson, 2006). This approach led to notable gains in core strength (ES=1.39) and performance (ES=2.16 vs. control). It aligns with findings that 1-3 minute rests enhance strength in multi-set resistance protocols (de Salles et al., 2009). Still, we acknowledge the potential endurance shortfall.

To enhance clarity, we will revise methods (Line 137–139): " A rest period of 1-2 minutes was provided between sets to support muscle recovery and balance strength-endurance adaptations for Tai Chi demands (20,21)."

In discussion: Combines Comments 2 and 5 into a single, concise sentence, replacing the original biomechanical sentence and rest interval insertion, aligning with paragraph's future research focus (Line�423–425)�"Future research might employ biomechanical tools and RPE scales to enhance monitoring of FTG stretch intensity and CTG endurance, ensuring protocol-specific data capture (24,29)."

Comment 3:

(Lines 143-144): The authors state: “The main stretching routine consisted of 4-5 stretches per session, held for 30-40 seconds each”. I miss some information about the amount of stretching tension supported by the participants as 30-40 sec is reported to be not enough to produce any effect on hamstrings elongation. Why don’t you apply longer stretching times? As is reported that stretching effects are better when the exposure times are longer, more than doing more than one set of shorter stretches.

Response:

We thank the reviewer for noting the hold durations of 30–40 seconds in the flexibility training group (FTG) protocol, including concerns about sufficiency for elongation and omitted tension details. We recognize these points merit explicit clarification to bolster reproducibility.

Durations were selected for athlete tolerability, emphasizing cumulative exposure (2–3 minutes per muscle group across 4–5 stretches), aligned with meta-analyses showing ROM improvements from accumulated volumes (Page, 2012). Tension was maintained at moderate discomfort, a standard safety cue, though not formally quantified.

We will revise methods (Line�143–145) to: " The routine included 4-5 stretches per session, held 30–40 seconds at moderate discomfort for tolerability and 2–3 minutes cumulative exposure per group (22). "

Comment 4:

Lines 147–149. The authors state: “Intensity was progressively increased bi-weekly by extending hold times or using assisted techniques, as needed. Participants rested for 30 seconds between stretches to prevent fatigue.” When did you apply assisted techniques or increased hold times? Justify the decision.

Response:

We appreciate the reviewer's call for details on progression timing and criteria in the flexibility training group (FTG) protocol. We admit the original text lacked sufficient specificity, impacting perceived rigor.

Progression was bi-weekly post-initial adaptation, decided via participant tolerance and coach observations, per progressive overload guidelines (American College of Sports Medicine, 2009). This ensured safe escalation, such as hold extensions or assisted techniques.

We will revise methods (Line�148–151) to: " Progression occurred bi-weekly after the initial adaptation phase, with decisions guided by participant tolerance (e.g., absence of excessive fatigue) and coach observations per overload guidelines (22,23). "

Comment 5:

Lines 155–157. The authors state: “The core group progressed through increased repetitions or resistance, while the flexibility group extended stretch durations”. Could you justify by previous studies the chosen method to increase the time exposure in stretches? Was Borg 0–10 scale also used for stretches? Did you consider any method to assess athletes’ perception of stretch intensity?

Response:

We thank the reviewer for querying the progression method, Borg RPE application, and perceived intensity assessments in the flexibility training group (FTG) protocol. We concede these details were under-elaborated, constituting a methodological limitation.

Bi-weekly duration increases followed progressive overload, supported by evidence of ROM gains from gradual volume escalation over ≥2 weeks (Konrad et al., 2024). RPE was not applied to stretches, relying on informal cues, unlike core sessions.

We will revise methods (Line�157–161) to: "with the core group increasing repetitions or resistance and the flexibility group extending stretch durations—supported by evidence of ROM gains from gradual volume escalations over ≥2 weeks (24). Session intensity was monitored via the Borg 0-10 RPE scale for the CTG only, recorded post-session for subjective workload (25) "

In discussion: Combines Comments 2 and 5 into a single, concise sentence, replacing the original biomechanical sentence and rest interval insertion, aligning with paragraph's future research focus (Line�423–424)�"Future research might employ biomechanical tools and RPE scales to enhance monitoring of FTG stretch intensity and CTG endurance, ensuring protocol-specific data capture (24,49).

Comment 6–9:

Table format should be improved (Tables 2–5).

Response:

We appreciate the reviewer’s suggestion. We have reformatted all tables according to PLOS ONE style guidelines, placing table titles above the tables, standardizing unit presentation, and aligning all values for improved clarity and readability.

Comment 10:

Lines 337–341. You should express the limitations on doing static stretches, as is highly demonstrated that static stretches reduce power and jump performance, so needed in martial arts.

Response:

We thank the reviewer for emphasizing the need to address limitations of static stretching in the discussion, particularly its acute reductions in power and jump performance, essential for martial arts. We acknowledge this as a valid point, as our flexibility training group (FTG) featured static-dominant elements, and omitting these effects could undervalue explanations for limited explosive outcomes.

Static stretching can impair immediate power and reactive strength, with meta-analyses indicating moderate decrements in jump height and force post-holds exceeding 60 seconds (Behm et al., 2016). Our FTG, while yielding range of motion gains (Sit and Reach, ES=1.44), showed minimal Rebound Jump Index or Jumping Lotus Kick improvements versus core training, likely influenced by these neuromuscular inhibitions (Warneke & Lohmann, 2024). This underscores stretching timing in dynamic disciplines, where post-training static approaches mitigate some risks but retain performance trade-offs.

To integrate this, we will revise the discussion (Line�401–403) to add: " This is particularly evident given static stretching’s acute reductions in power and jump performance, which likely limited FTG’s explosive outcomes, underscoring the need for dynamic approaches (45,46)"

Comment 11:

Lines 372–374. Not all core stability exercises have this effect. Proprioception highly depends on imagery, focusing attention and reduction of competing afferent inputs. Be careful on comparing proprioception with balance performance.

Response:

We appreciate the reviewer's nuanced point on core stability exercises' effects on proprioception, underscoring its reliance on imagery, attention, and afferent modulation. We concede this risks conflating proprioception with balance, a distinction essential for accurate interpretation.

Proprioception involves sensory integration, while balance is a functional output; core training may enhance the latter without uniformly affecting the former, as Hrysomallis (2011) notes in reviewing balance's role in athletic performance, where core interventions improve stability via neuromuscular pathways but vary in proprioceptive impact.

To refine, we will revise discussion (Lines�375–377): " Core stability training enhances neuromuscular balance control, a critical factor for rotational shifts, though often conflated with proprioception, which involves sensory integration (38)"

Once again, we thank the reviewer for their thorough evaluation and constructive comments, which have greatly improved the quality and clarity of our manuscript.

---

## [Decision Letter · Decision Letter 1]

13 Oct 2025

Core Training Elicits Greater Improvements Than Flexibility Training in Jumping Lotus Kick Performance and Physical Attributes of Tai Chi Athletes: A Randomized Controlled Trial

PONE-D-24-44478R1

Dear Dr. Gong,

We’re pleased to inform you that your manuscript has been judged scientifically suitable for publication and will be formally accepted for publication once it meets all outstanding technical requirements.

Kind regards,

Maheshkumar Baladaniya

Academic Editor

PLOS ONE

Additional Editor Comments (optional):

Manuscript is in acceptable form for the publication.

Reviewers' comments:

Reviewer's Responses to Questions

**Comments to the Author**

Reviewer #2: All comments have been addressed

Reviewer #3: All comments have been addressed

2. Is the manuscript technically sound, and do the data support the conclusions?

Reviewer #2: Yes

Reviewer #3: Yes

3. Has the statistical analysis been performed appropriately and rigorously?

Reviewer #2: Yes

Reviewer #3: Yes

4. Have the authors made all data underlying the findings in their manuscript fully available?

Reviewer #2: Yes

Reviewer #3: Yes

5. Is the manuscript presented in an intelligible fashion and written in standard English?

Reviewer #2: Yes

Reviewer #3: Yes

Reviewer #2: The manuscript is now organized. The theoretical framework and statistical analyses are articulated clearly, and the discussion offers significant implications for both research and practice. I think this research offers a significant addition to the body of work on digital HRM, knowledge sharing, and organizational innovation.

Reviewer #3: The authors have adequately addressed all comments raised by the reviewers. The current manuscript presents valuable findings; however, it lacks integration of a multidisciplinary perspective that could significantly enhance its scientific depth and translational relevance. For instance, the discussion focuses primarily on the physiological aspects without sufficient linkage to clinical, psychological, or public health implications. Incorporating reference from studies that explore the intersection between physical performance, mental health, and preventive medicine would strengthen the rationale and demonstrate a broader contextual understanding. Add the phrase like below in discussion section..

“Beyond musculoskeletal improvements, structured exercise regimens have been shown to foster psychological resilience and enhance mind–body coordination [Nasif et al., 2025 https://doi.org/10.31579/2578-8868/359 ]. Such holistic adaptations are central to performance in Tai Chi, where mental composure and neuromuscular precision are intertwined.”

**Do you want your identity to be public for this peer review?** For information about this choice, including consent withdrawal, please see our Privacy Policy

Reviewer #2: No

Reviewer #3: No

---

## [Editor Report · Acceptance letter]

PONE-D-24-44478R1

PLOS One

Dear Dr. Gong,

I'm pleased to inform you that your manuscript has been deemed suitable for publication in PLOS One. Congratulations! Your manuscript is now being handed over to our production team.

Kind regards,

on behalf of

Dr. Maheshkumar Baladaniya

Academic Editor

PLOS One